

# Targeted genomic enrichment and sequencing of CyHV-3 from carp tissues confirms low nucleotide diversity and mixed genotype infections

Saliha Hammoumi[1], Tatiana Vallaeys[2], Ayi Santika[3], Philippe Leleux[4], Ewa Borzym[5], Christophe Klopp[4] and Jean-Christophe Avarre[1]

[1] Institut des Sciences de l'Evolution de Montpellier, UMR226 IRD-CNRS-UM-EPHE, Montpellier, France
[2] Université de Montpellier, Montpellier, France
[3] Main Center for Freshwater Aquaculture Development, Sukabumi, Indonesia
[4] Plate-forme Genotoul Bioinfo, UR875 Biométrie et Intelligence Artificielle, Institut National de la Recherche Agronomique, Castanet-Tolosan, France
[5] Department of Fish Diseases, National Veterinary Research Institute, Pulawy, Poland

Corresponding author
Jean-Christophe Avarre,
jean-christophe.avarre@ird.fr

## ABSTRACT

Koi herpesvirus disease (KHVD) is an emerging disease that causes mass mortality in koi and common carp, *Cyprinus carpio* L. Its causative agent is Cyprinid herpesvirus 3 (CyHV-3), also known as koi herpesvirus (KHV). Although data on the pathogenesis of this deadly virus is relatively abundant in the literature, still little is known about its genomic diversity and about the molecular mechanisms that lead to such a high virulence. In this context, we developed a new strategy for sequencing full-length CyHV-3 genomes directly from infected fish tissues. Total genomic DNA extracted from carp gill tissue was specifically enriched with CyHV-3 sequences through hybridization to a set of nearly 2 million overlapping probes designed to cover the entire genome length, using KHV-J sequence (GenBank accession number AP008984) as reference. Applied to 7 CyHV-3 specimens from Poland and Indonesia, this targeted genomic enrichment enabled recovery of the full genomes with >99.9% reference coverage. The enrichment rate was directly correlated to the estimated number of viral copies contained in the DNA extracts used for library preparation, which varied between ~5000 and ~$2 \times 10^7$. The average sequencing depth was >200 for all samples, thus allowing the search for variants with high confidence. Sequence analyses highlighted a significant proportion of intra-specimen sequence heterogeneity, suggesting the presence of mixed infections in all investigated fish. They also showed that inter-specimen genetic diversity at the genome scale was very low (>99.95% of sequence identity). By enabling full genome comparisons directly from infected fish tissues, this new method will be valuable to trace outbreaks rapidly and at a reasonable cost, and in turn to understand the transmission routes of CyHV-3.

Subjects Aquaculture, Fisheries and Fish Science, Marine Biology, Veterinary Medicine
Keywords Cyprinid herpesvirus 3, Genome, Targeted genomic enrichment, Mixed infection, Variability
## INTRODUCTION

Koi herpesvirus disease (KHVD) is an emerging disease that causes mass mortalities in koi and common carp, *Cyprinus carpio* L. Since the first report in the late 1990s (*Hedrick et al., 2000*), the disease has spread to many countries worldwide (*Pokorova et al., 2005*). The causative agent, cyprinid herpesvirus 3 (CyHV-3), is a large double-stranded DNA virus belonging to the *Alloherpesviridae* family, genus *Cyprinivirus* (*Davison, 2010*; *King et al., 2011*; *Waltzek et al., 2005*). Twelve alloherpesviruses have been described to date: two of them infect amphibians while the ten remaining infect various fish species, including eel, salmon, catfish, goldfish and sturgeon (*Waltzek et al., 2009*).

Although the pathogenesis of this virus is well understood and well described in the literature, little is known about the genomic diversity and evolution. Sequencing of three complete CyHV-3 genomes of isolates originating from the USA (U), Israel (I) and Japan (J) (*Aoki et al., 2007*) revealed that sequences were highly similar (>99.9%), which was consistent with a scenario of a unique virus that spread worldwide. Meanwhile, accurate genomic comparisons highlighted two main genetic lineages formed by the U/I and J strains. The existence of two lineages was further confirmed on a larger set of European and Asian isolates by using PCR-based molecular markers targeting various regions of the genome (*Bigarré et al., 2009*; *Kurita et al., 2009*). These two lineages were designated as European and Asian. Finally, the use of variable number of tandem repeats (VNTR) revealed a whole range of variants associated with both European and Asian lineages (*Avarre et al., 2011*). It was also demonstrated that genetically distinct strains of CyHV-3 may co-occur in a single enzootic area, and that carp were frequently infected with a mix of genotypes (*Avarre et al., 2012*; *Sunarto et al., 2011*). Although viruses evolve more rapidly than their hosts, the differentiation ability of CyHV-3 remains mostly unknown, and consequently, it is difficult to predict whether the presence of distinct genotypes within a single host results from distinct infection events, from the evolution of a unique strain, or from both. Thus, characterizing these genetic variants at the genome level is critical to better understand the ecological and evolutionary significance of such mixed infections, especially with regards to the virulence and the modes of propagation of the virus.

In this context, the objective of the present work was to develop a method enabling the sequencing of CyHV-3 full-length genomes directly from their host. Targeted genomic enrichment (TGE), or targeted sequence capture prior to sequencing, has been described for a large panel of applications (*Mertes et al., 2011*). In spite of its great potential in virology, the application of this approach has remained limited since its first description and successful application to human herpesviruses (*Depledge et al., 2011*). Using a set of probes designed to specifically capture CyHV-3 sequences, seven genomes were fully sequenced from carp gill tissue samples collected during several outbreaks. In order to reduce the cost associated with TGE, the influence of pre-capture multiplexing on the ability to recover full genomes and to discover mutations in CyHV-3 with high confidence was also evaluated.

**Table 1   Assay design for CyHV-3 sequence capture.**

| Assay format | Specimen name | Estimated number of viral copies[a] | Country and year of collection | Reference | Experiment accession # |
|---|---|---|---|---|---|
| Simplex | PoB3_1 | 4.8E+3 | Poland, 2013 | This study | SRX1071741 |
| Simplex, double capture | PoB3_2 | 4.8E+3 | Poland, 2013 | This study | SRX1071742 |
| 4-plex | PoB3_3 | 4.8E+3 | Poland, 2013 | This study | SRX1071743 |
|  | J1_101110 | 1.5E+7 | Indonesia, 2010 | Avarre et al. (2012) | SRX1071745 |
|  | CB4_181110 | 1.9E+7 | Indonesia, 2010 | Avarre et al. (2012) | SRX1071746 |
|  | PP3_070411 | 5.7E+6 | Indonesia, 2011 | Avarre et al. (2012) | SRX1071747 |
| 4-plex | PoB3_4 | 4.8E+3 | Poland, 2013 | This study | SRX1071744 |
|  | I_10-3 | 1.5E+5 | Indonesia, 2010 | Avarre et al. (2011) | SRX1071748 |
|  | I_09-2i3 | 1.5E+5 | Indonesia, 2009 | Avarre et al. (2011) | SRX1071749 |
|  | J2_101110 | 8.8E+4 | Indonesia, 2010 | Avarre et al. (2012) | SRX1071750 |

**Notes.**
[a]Before capture of viral DNA.

## MATERIAL AND METHODS

### CyHV-3 specimens and DNA extraction

Seven CyHV-3 specimens were used in the present study. They consisted of gill tissues collected from infected carps and stored in 96% EtOH at $-20\,°C$. Six of them (J1_101110, CB4_181110, PP3_070411, I_10-3, I_09-2i3, J2_101110) were obtained from moribund cultured common carps collected in Indonesia (Sukabumi district, West Java), during different outbreaks (Table 1), whereas the seventh (PoB3) was obtained from a common carp collected during an outbreak in Poland. Total genomic DNA was extracted from a small gill fragment with the DNAzol kit (Life technologies) and the Wizard Genomic DNA purification kit (Promega) for the Indonesian and Polish specimens, respectively, following the appropriate protocols provided by the manufacturers. DNA concentration was measured by fluorometry (Qbit, Life Technologies), and its integrity was verified by capillary electrophoresis (Bioanalyzer 2100; Agilent). The viral DNA concentration was determined in each sample by real-time PCR, using the same method as previously described (*Avarre et al., 2011*; *Yuasa et al., 2005*).

### CyHV-3 genome enrichment and sequencing

About 2 μg of total DNA were used to establish the genomic libraries of each sample, using the TruSeq DNA sample preparation kit (Illumina). DNA was sonicated using a Bioruptor device (Diagenode). The average insert size used for sequencing was ∼300 bp. After adapter ligation using the Truseq kit, libraries were subjected to a viral-specific sequence capture using the SeqCap EZ library SR technology (Roche-Nimblegen). A large set of 50–105 mer CyHV-3 specific probes was designed (Roche-Nimblegen) to theoretically capture all DNA sharing more than 90% identity with the published CyHV-3 genome of strain J (GenBank #AP008984). This probe set (Roche Diagnostics design OID 41307, IRN 4000012980; Roche) is available upon demand.
A pre-capture ligation-mediated PCR (LM-PCR) consisting of 8 cycles of PCR was first performed according to the manufacturer's instructions (Illumina). After purification using Agencourt Ampure XP beads (Beckman Coulter), the amplified products were hybridized with the CyHV-3 probes at 47 °C for 64 h. In 4-plex assays, 250 ng of four amplified products were pooled prior to hybridization. At the end of incubation, hybridized DNA was recovered using Nimblegen capture beads. Then 18 cycles of a post-capture LM-PCR were carried out on each captured DNA sample. DNA was again purified using Ampure beads, and its concentration was measured using a spectrophotometer (Nanodrop 1000) while its integrity was verified using a Bioanalyzer 2100 (Agilent).

In view of reducing the experimental costs, we compared the efficiency of a pre-capture multiplexing (4-plex) design to that of a simplex one, using specimen PoB3 as "control" because this sample displayed the lowest viral load (Table 1). Each library had initially been ligated with a unique Illumina adaptor, and two 4-plex assays were designed according to the estimated viral loads: the first included PoB3 and 3 Indonesian specimens with high viral loads ($>10^6$ genome copies) whereas the second comprised PoB3 and the 3 remaining Indonesian specimens that displayed lower loads ($\leq 1.5 \times 10^5$ genome copies). In parallel, PoB3 was also subjected to a double-capture protocol, which consisted of two overnight hybridizations at 47 °C followed by a non-specific DNA elution using capture beads. All sample combinations were sequenced on a single lane of a HiSeq2000 platform (Illumina) in a paired-end (100 bases) format, at Montpellier Genomix (Montpellier, France).

## Genome assembly/mapping and detection of variations

Raw sequences (fastq files) were stored in the public Sequence Read Archive (SRA) repository under the SRP study accession SRP059764 as well as in the NG6 repository (*Mariette et al., 2012*) of GenoToul (Toulouse, France) for analysis purposes. A first quality control consisted in the visual inspection of the different graphs produced by fastqc (http://www.bioinformatics.babraham.ac.uk/projects/fastqc/), as well as in the alignment of the reads against classical contaminants such as *Escherichia coli*, yeast and phages. All the raw reads which passed this quality control were aligned to the CyHV-3 J strain genome (accession number AP008984) using BWA MEM (version bwa-0.7.5a) (*Li & Durbin, 2009*). Low quality alignments, i.e., reads aligning more than once or with poor sequencing quality inducing alignment quality lower than 30, were filtered out. The read depth was computed for the reference J-strain genome using Samtools (version 1.1) (*Li et al., 2009*). Variant calling was performed on the filtered alignments using the Genome Analyses Tool Kit (GATK, version 3.3.0) (*DePristo et al., 2011*), which improves the variant calling process through read realignment and base quality recalibration. PCR duplicates were filtered out using picard tools MarkDuplicates (version 1.88). Variations were thus obtained for single nucleotide polymorphisms (SNPs), insertions and deletions (indels) in VCF format files, using a Q20 filter. Both mapping and variant calling results were manually checked using the Integrative Genomics Viewer (IGV, version 2.3) (*Thorvaldsdottir, Robinson & Mesirov, 2013*). Finally, regions showing high coverage were checked for contamination using blastn with standard settings (*Altschul et al., 1997*) against the NCBI nucleotide collection (nr/nt)

database. Figures representing the sequencing depth and the number of small nucleotide variants were created with the R packages ggplot2 and limma, respectively.

A *de novo* assembly of the reads was concomitantly performed for each sample using ABySS, and Cap3 programs. Briefly, fastq files were trimmed with Fastq trimmer (version 1.0.0) (*Blankenberg et al., 2010*) using the default parameters and the resulting reads were assembled in paired-end by ABySS v1.5.0 (*Simpson et al., 2009*). The obtained contigs were subjected to a new assembly using Cap3 (*Huang & Madan, 1999*), and resulting assemblies were mapped on a CyHV-3 reference genome (strain U, accession number DQ657948) using Bowtie 2 (version 2.2.1) program (*Langmead et al., 2009*). Consensus sequences were generated from these assemblies, in which indels and SNPs were validated when more than 50% of the reads harbored the variation.

## Genome comparisons

Read alignments were also subjected to a genetic distance analysis. Variant calling files of individual specimens were compared through an identity-by-state calculation using PLINK, with the distance option (*Chang et al., 2015*). Reads were artificially created for the three reference genomes U, I and J in order to include them in the analyses. For this purpose, 100-base reads were artificially produced every five bases along the reference genome with a simple python script. These reads were then randomly shuffled to form a set of reads with an even 20× depth, using samtools faidx (version 0.1.19). Then a hierarchical cluster analysis was computed with a complete-linkage clustering method, using the standard parameters, and dendrograms were plotted with the hclust R package. A visual representation of the distribution of nucleotide variants (SNPs + indels) was also created with Circos software (*Krzywinski et al., 2009*), using the genome of CyHV-3 J strain as reference. For clarity, only the genomes of PoB3_1, J1_101110 and I09_2i3 were represented.

The seven consensus sequences were aligned together with the three already sequenced genomes (J, U and I strains) using the online Multiple alignment program for amino acid or nucleotide sequences (Mafft program version 7, http://mafft.cbrc.jp/alignment/server/). The resulting alignments were visualized and manually corrected with GeneDoc version 2.7 (*Nicholas, Nicholas & Deerfield, 1997*). Then the evolutionary history was inferred by the Neighbor-Joining method (*Saitou & Nei, 1987*), and evolutionary distances were computed with MEGA6 (*Tamura et al., 2013*), using the Maximum Composite Likelihood method (*Tamura, Nei & Kumar, 2004*). All positions containing gaps and missing data were eliminated.

## Nucleotide sequence accession numbers

The raw sequences (fastq files) of the whole project are available under the SRP study accession SRP059764 (http://trace.ncbi.nlm.nih.gov/Traces/sra/?study=SRP059764). It is organized in 7 biosamples and 10 experiments, as shown in Table 1. Sequences corresponding to the 7 consensus genomes have also been submitted to GenBank and can be accessed under the following accession numbers: KX544842, KX544843, KX544844, KX544845, KX544846, KX544847 and KX544848 (for PoB3, J2_101110, CB4_181110, J1_101110, I_09_2i3, I_10_3 and PP3_070411, respectively).

**Table 2  Main features of enrichment and sequencing results.**

| Specimen name | # raw reads | % mapped reads[a] | % mapped reads after duplicate removal[a] | Mean depth (x)[a,b] | Coverage along the reference genome (%)[a] | Number of uncovered positions[a] | Estimated number of viral copies |
|---|---|---|---|---|---|---|---|
| PoB3_1 | 22,852,739 | 28.73 | 3.22 | 221 | 99.99 | 32 | 4.8E+3 |
| PoB3_2 | 28,611,191 | 89.40 | 4.59 | 409 | 99.05 | 2,600 | 4.8E+3 |
| PoB3_3 | 9,163,662 | 16.67 | 7.22 | 219 | 99.92 | 218 | 4.8E+3 |
| J1_101110 | 32,892,125 | 80.47 | 49.66 | 5,931 | 99.94 | 168 | 1.5E+7 |
| CB4_181110 | 49,788,380 | 88.81 | 50.70 | 9,179 | 99.99 | 29 | 1.9E+7 |
| PP3_070411 | 42,335,046 | 76.32 | 43.99 | 6,741 | 99.96 | 115 | 5.7E+6 |
| PoB3_4 | 16,182,800 | 33.78 | 4.90 | 246 | 100 | 0 | 4.8E+3 |
| I_10_3 | 44,364,454 | 67.84 | 16.46 | 2,601 | 100 | 0 | 1.5E+5 |
| I_09_2i3 | 56,046,694 | 74.00 | 19.87 | 4,015 | 100 | 0 | 1.5E+5 |
| J2_101110 | 14,067,420 | 31.61 | 5.26 | 235 | 99.90 | 272 | 8.8E+4 |

**Notes.**
[a]Only Q30 sequences were considered.
[b]After duplicate removal.

## RESULTS

### Viral genomic DNA enrichment: read mapping, sequencing depth and genome coverage

A total of 316,007,054 reads of 100 bp were obtained for the ten libraries, resulting in about 31.6 Gb of sequence data. The number of reads per library ranged from 9,163,662 (PoB3_3) to 56,046,694 (I_09-2i3) (Table 2). The rate of enrichment, reflected by the proportion of mapped reads, apparently correlated with the initial viral load, as it tended to increase concomitantly. For specimens with viral loads $>10^6$ copies, the proportion of mapped reads (after duplicate removal) was comprised between 43.99% (PP3_070411) and 50.70% (CB4_181110). Conversely, for PoB3, it varied between 3.22% and 7.22% of the total number of reads, depending on the assay format. Removal of duplicate sequences generally led to a significant reduction in the proportion of reads that could be mapped on the CyHV-3 reference genome (strain J). Sequence depth along this reference genome was variable both across and within specimens. It was $>2,000\times$ on average for all specimens with an estimated number of viral copies $>10^5$, and $>200\times$ for all other specimens (Table 2 and Fig. S1). For most samples, there was an over representation of reads at positions 226,373–226,831 and 228,015–228,163 (according to KHV-J coordinates, GenBank # AP008984). A Blast search revealed that these regions shared 97% and 94% identity with sequences of *Cyprinus carpio* genome (GenBank # LN591727 and LN591823, respectively). However, this over representation was attenuated in samples with high initial viral loads (J1_101110, CB4_181110 and PP3_070411). A higher coverage ($\times30$) was also noticed for J2_101110 specimen in the region corresponding to positions 45,500–47,700; yet this portion of CyHV-3 genome did not return any significant blast hit. However, since the genome of *Cyprinus carpio* is not yet complete, it cannot be ruled out that these sequences belong to the carp genome. In spite of these slight differences, genomes could
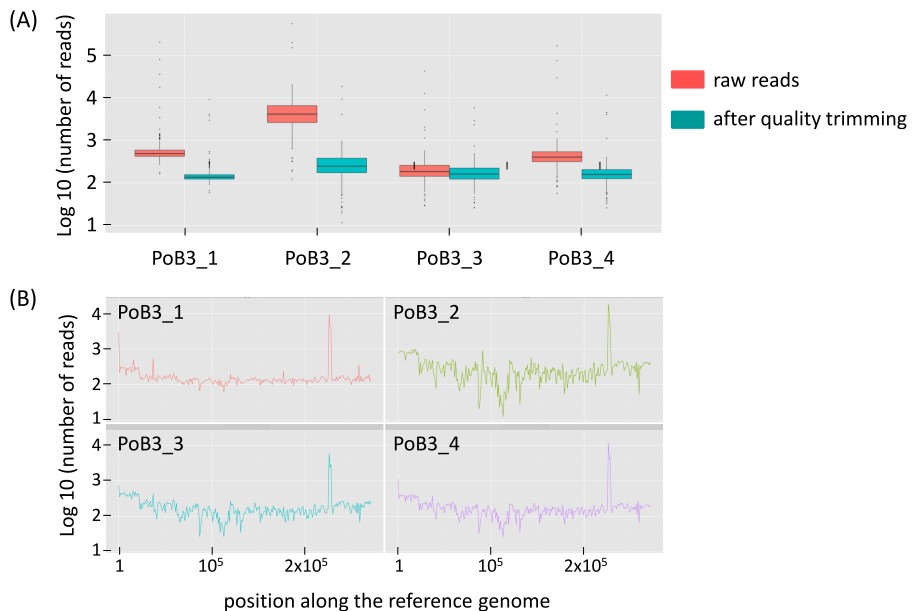

**Figure 1** **Sequencing depth for PoB3 samples.** (A) Boxplot representing the depth distribution in log10 of raw reads (red) and filtered reads (blue), i.e., after Q30 mapping quality trimming and duplicate removal. (B) Per base sequencing depth along the KHV-J reference genome. The figure was created with the R package ggplot2.

be recovered in their totality (or near-totality) from all samples, since coverage along the reference sequence was above 99.9%, with the exception of PoB3_2 for which coverage was slightly lower (99.05%).

## Comparison of PoB3 sequences according to the assay format

Comparison of PoB3 in the different situations suggests that pre-capture multiplexing had an influence on the number of raw reads: the number of obtained sequences was highest when PoB3 was enriched separately, lowest when mixed with specimens with high viral loads (PoB3_3) and intermediate when mixed with specimens displaying lower viral loads (PoB3_4). The proportion of mapped reads (before duplicate removal) followed the same trend, with the noticeable exception of PoB3_2 (89.40%) (Table 2 and Fig. S2). However, this high proportion of mapped reads obtained with the double-capture protocol (PoB3_2) dropped to 4.59% after sequence duplicate removal, suggesting that the majority of these sequences corresponded to duplicates. Even though the mean depth values of the 4 PoB3 samples were comparable, their distribution along the genome, however, displayed marked differences: whereas sequence representation was the steadiest along the genome for PoB3_1, it was uneven for PoB3_2, and in a lesser extent for PoB3_3 and PoB3_4 (Fig. 1).

When compared to KHV-J reference strain, the number of variations (SNP + indel) observed for the 4 PoB3 samples varied from 423 (PoB3_1) to 457 (PoB3_3). Among these variations, 335 were shared by all 4 PoB3 samples (Fig. 2). PoB3_3 displayed the highest number of 'unique' variations (75), representing ∼16% of the total variations recorded for this sample. Examination of these 75 positions revealed that they all were 'heterozygous,'
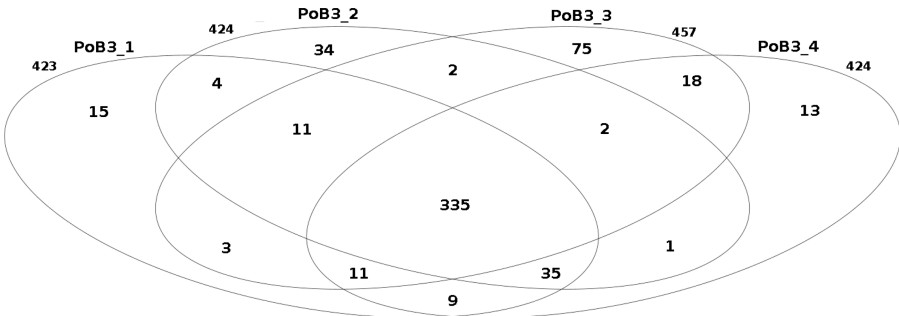

**Figure 2** **Comparison of the number of small nucleotide variants between PoB3 samples.** Intersections represent the variants that are shared between samples. This Venn diagram was created with the R package limma.

**Table 3** **Number of single nucleotide polymorphisms (SNPs) and insertions/deletions (indels) in the different specimens compared to KHV-J.**

| Specimen | SNP | Indel | % small nucleotide variants [a] | % intra-specimen heterogeneous sites[a,b] |
|---|---|---|---|---|
| PoB3_1 | 215 | 208 | 0.14 | 31.8 |
| PoB3_2 | 216 | 208 | 0.14 | 33.6 |
| PoB3_3 | 230 | 227 | 0.15 | 95.4 |
| J1-101110 | 47 | 66 | 0.04 | 31.3 |
| CB4-181110 | 43 | 57 | 0.03 | 31.3 |
| PP3_070411 | 66 | 60 | 0.04 | 42.5 |
| PoB3_4 | 213 | 211 | 0.14 | 93.0 |
| I10-3 | 72 | 53 | 0.04 | 72.1 |
| I_09-2i3 | 60 | 56 | 0.04 | 54.6 |
| J2_101110 | 94 | 77 | 0.06 | 81.9 |

**Notes.**
[a] Cut-off value set at 10% of the total number of reads at each specific position.
[a] Excluding the terminal repeats.

i.e., they consisted in a mix of two haplotypes, including that of the reference genome. Fifty five of these variations were shared with at least one of the three other co-enriched samples (J1-101110, CB4-181110, PP3_070411), and 10 additional ones were located in the region that matches with the host genome (between positions 226,373 and 228,163 according to KHV-J coordinates). A distance analysis including the four PoB3 replicates (Fig. S3) indicated that the impact of these differences on the general phylogeny was very limited.

## Inter- and intra-specimen variations

Using KHV-J as reference, sequence identity ranged from 99.85% to 99.97%. The most divergent sequence was that of PoB3. This corresponded to approximately 425 sites that varied from the reference KHV-J sequence. The proportion of single nucleotide polymorphisms (SNP) and insertions/deletions (indel) was nearly 50%–50% for all samples (Table 3). As can be visualized on Fig. 3, variations were distributed across the

entire genome, with no particular hotspot of variability. In addition to the differences between specimens, sequence variability also occurred within specimens. Indeed, the proportion of varying sites harboring more than one allele (using the GATK criteria) was above 30% for all specimens (Table 3). If some of this intra-specimen variability may probably be attributed to experimental biases (see 'Discussion' below), it nonetheless suggests the presence of multiple genotypes within a single specimen. Because of this intra-specimen sequence heterogeneity, distance analysis between specimens was realized directly from vcf files, using the PLINK software. Results indicated that PoB3 belonged to the European lineage, whereas all Indonesian specimens belonged to the Asian one (Fig. 4). Multiple sequence alignment realized on the seven generated consensus genomes and the three reference CyHV-3 genomes led to the same result (Fig. S4).

## DISCUSSION

Concentration of viral particles through cell culture propagation (providing a suitable cell line is available for the investigated virus) is not always successful and requires days to obtain sufficient viral titers. Using targeted sequence capture prior to sequencing enabled here to recover ten complete (or near-complete) 295-kbp genomes in one single run of sequencing, directly from infected tissues, and with mean read depth varying from $219\times$ to $9179\times$. For comparison, *Liu et al.(2011)* used 24 genomic DNA libraries of Epstein Barr virus (EBV) to recover one full genome from a tumor tissue with an average read depth of $17\times$. This illustrates the potential of TGE to overcome the high discrepancy in the proportions of cellular and viral genomic DNA in raw samples obtained from infected tissues. Sample multiplexing is now routinely employed in next-generation sequencing, through the use of specific barcodes (*Parameswaran et al., 2007*). However, there are a limited number of studies describing pre-capture multiplexing, though it may improve both efficiency and cost-effectiveness of TGE (*Shearer et al., 2012*). In the present case, a rough estimation indicated that $4\times$ multiplexing resulted in a drop of the per sample price from ~1,600€ to ~450€ (excluding sequencing costs).

Although there are no replicates to allow for statistical confirmation, the results obtained on PoB3 samples revealed some interesting differences. First, the double-capture protocol resulted in a higher number of uncovered positions, suggesting that the repeated amplifications have generated a bias in sequence representation. Though this aspect is not well documented in the literature, it has been shown, however, that the time of barcode incorporation during the library preparation and its distance from the priming site, as well as the PCR extension temperature, were potential sources of bias (*López-Barragán et al., 2011*; *Van Nieuwerburgh et al., 2011*). Second, while pre-capture multiplexing entailed a decrease in the number of reads, it did not compromise the sequencing depth or the coverage along the genome. Third, there were slight differences in the number of variations found between the four PoB3 samples and KHV-J reference genome, with PoB3_3 displaying the highest number of 'unshared' variations, i.e., variations that were not recorded in the other PoB3 samples. It is interesting to note that the 75 variations that were specific to PoB3_3 were all heterozygous, and that the majority of them were shared by at least one

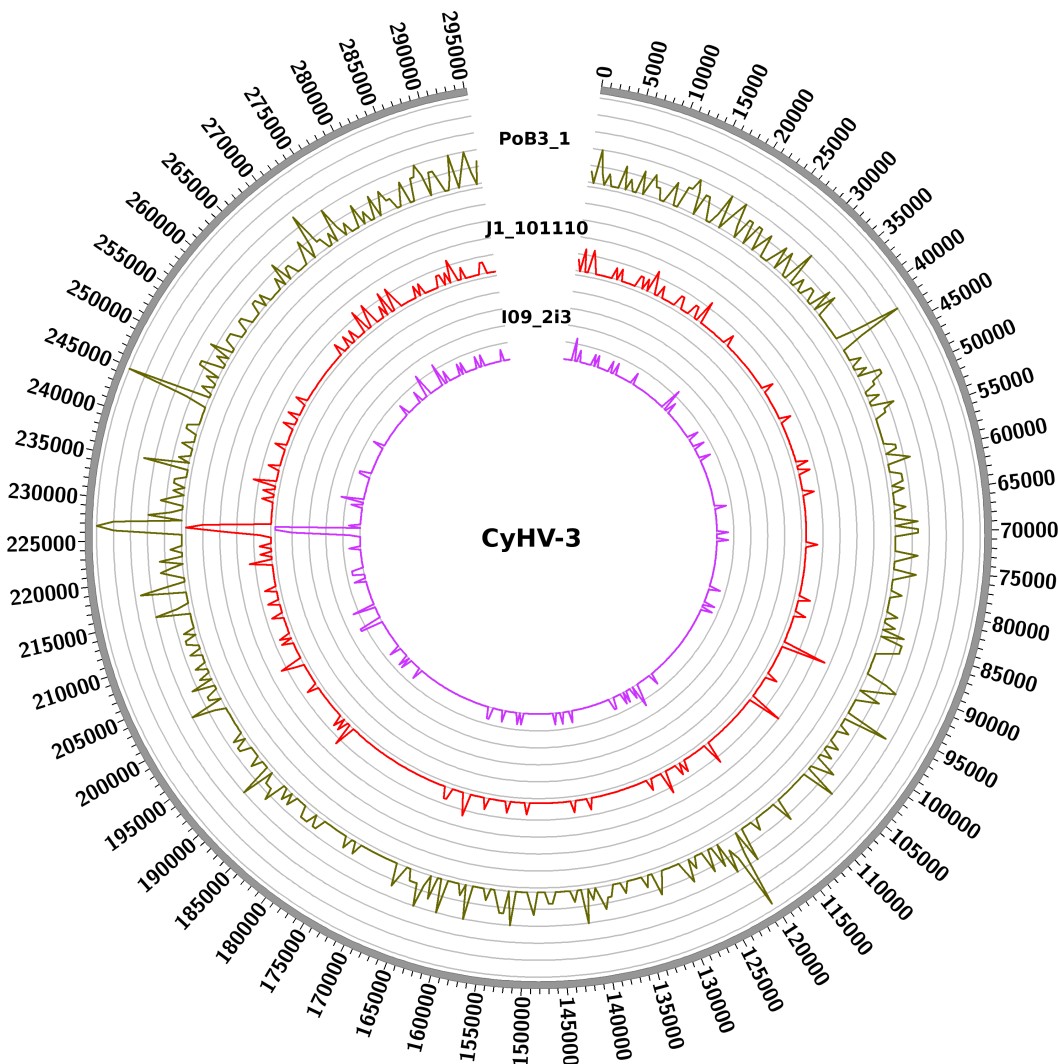

**Figure 3  Distribution of nucleotide variations along CyHV-3 genomes.** Numbers correspond to the genomic coordinates of CyHV-3 J strain, and the internal curves depict the small nucleotide variants (including SNPs and indels) found in the three genomes (PoB3_1, J1_101110 and I09_2i3) compared to CyHV-3 J strain. Peak height is proportional to the number of variations in a given region. A window size of 5 kb was used.

of its co-enriched counterparts. Of these 75 variants, only 10 were located in the region that shares a high sequence identity with the carp reference genome. This evidence is likely indicative of low level cross-contamination with co-enriched samples. Comparison with PoB3_4, for which the 13 specific variations were also heterozygous and shared by at least one of the multiplexed samples, suggests that such a contamination would be favored by the difference in the initial viral loads of the multiplexed samples. For this reason, it seems advisable to multiplex samples with comparable viral loads.

It is generally acknowledged that errors during the mapping of cluster coordinates may result in 1–2% of bias in assignment of reads to sample during demultiplexing (*Krueger, Andrews & Osborne, 2011*). Considering the great difference in the sequencing

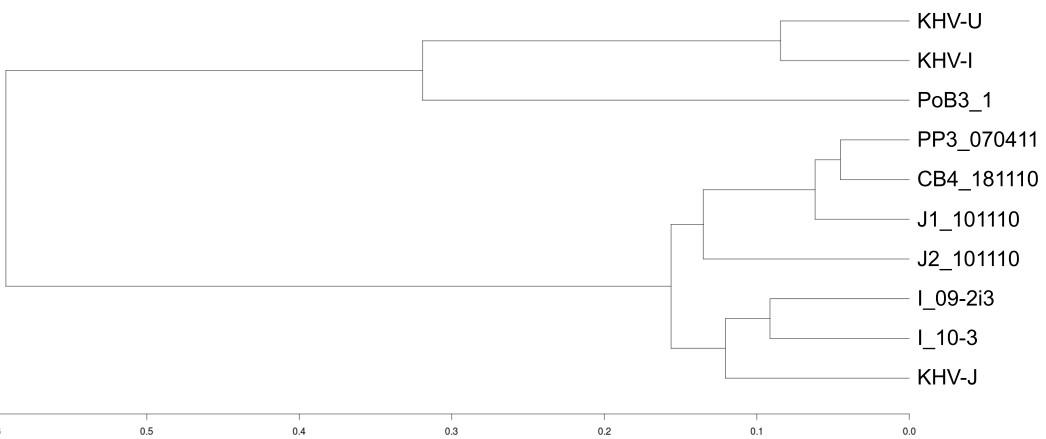

**Figure 4  Genetic distances between CyHV-3 specimens.** A distance matrix between samples was first computed using an Identity By State (IBS) distance. Specimens were then grouped into clusters using the R package hclust and the corresponding dendrogram plotted. The scale represents the maximum IBS-distance between two elements of linked clusters (branches in the tree).

depth between PoB3 samples and the other specimens, such a "contamination" could explain these faint differences between PoB3 samples. Moreover, an additional source of inaccuracy may also emanate from amplification and/or sequencing errors. In order to evaluate the contribution of pre-capture multiplexing on these observed differences, a specimen with a high viral load ($1.5 \times 10^8$ viral copies) was passaged on CCB (common carp brain) cells and sequenced: (1) after TGE in a 4-plex format with specimens of comparable viral loads; (2) without enrichment. Comparison of the resulting sequences revealed only 18 differences over the 295-kbp genomes between the enriched and non-enriched samples (data concerning this specimen and additional ones will be published separately). This result brings additional evidence that pre-capture multiplexing does not induce any significant bias as long as the multiplexed specimens have comparable viral loads, and/or as long as the initial viral amount is sufficient to generate elevated read depth ($>1,000\times$). In any case, the differences between the four PoB3 samples represented nearly 0.004–0.025% of the genome, i.e., 4–25 differences every 100 kb. This error rate appears acceptable, especially if we consider the low number of viral copies that were initially contained in this specimen.

Considering the relatively low viral load of J2_101110 specimen, we can also expect that a number of observed variations may constitute 'false callings'. This can be anticipated by (i) the total number of detected variations (171), which is slightly higher than in the other Indonesian specimens (100–126); and (ii) the higher proportion of heterozygous variations compared with the other Indonesian specimens. Even though the proportion of heterozygous positions is certainly over-estimated, especially in samples that were multiplexed prior to TGE, such a high rate of intra-specimen variability confirms previous findings that carp are frequently infected with a mix of CyHV-3 genotypes (*Sunarto et al., 2011*). The way such mixed infections arise in their host still remains speculative, as well as their ecological and evolutionary advantage. Now, the possibility

to access full viral genomes from their host will undoubtedly contribute to a better understanding of CyHV-3 mixed infections.

The genomic distance analysis indicated that PoB3 specimen belonged to the European lineage, whereas the Indonesian samples fell in the Asian lineage. However, as was formerly shown (*Aoki et al., 2007*; *Li et al., 2015*), the divergence remains very low at the genome scale. The first KHVD outbreak reported in Indonesia dates back to 2002 (*Sunarto et al., 2002*). Since then, the disease rapidly spread throughout the whole country, especially in the West Java region (*Sunarto, Rukyani & Itami, 2005*). The use of 12 VNTR (variable number of tandem repeat) markers on the six investigated CyHV-3 Indonesian specimens revealed an important diversity of genotypes, which is not reflected at the genome scale (*Avarre et al., 2012*). This is not surprising, as tandem repeats evolve faster than any other part of a genome. Whereas some tandem repeats do not carry any adaptive value, others, however, allow functional diversification and may be involved in pathogenic pathways (*Mrázek, Guo & Shah, 2007*; *Treangen et al., 2009*). Therefore, full genomic comparisons will probably be more suitable to trace back outbreaks and to understand the transmission routes of CyHV-3, as they will also enable the study of the variations of short tandem repeats contained in the genomes of CyHV-3.

## CONCLUSION

This TGE strategy enabled the successful recovery of full-length genomes from samples containing as little as 5,000 viral copies. Though pre-capture multiplexing of samples with low viral loads is prone to generate false variants, it dramatically reduces the cost of an experiment. From our results, we would advise to pre-capture multiplex only samples with an initial number of viral copies above $10^5$, and to multiplex samples with comparable loads whenever possible. This TEG strategy will enable to analyze the genomic diversity of CyHV-3 and to study the dynamics of *in vivo* infections, by accessing the possible heterogeneity in genome populations through the analysis of the nucleotide composition at each position along the genome.

## ACKNOWLEDGEMENTS

This is publication IRD-DIVA-ISEM 2016-162.

### Funding

This work was supported by the ERA-NET EMIDA funded project MOLTRAQ (molecular tracing of viral pathogens in aquaculture). The funders had no role in study design, data collection and analysis, decision to publish, or preparation of the manuscript.

### Grant Disclosures

The following grant information was disclosed by the authors:
ERA-NET EMIDA.

## Competing Interests

The authors declare there are no competing interests.

## Author Contributions

- Saliha Hammoumi conceived and designed the experiments, performed the experiments, analyzed the data, wrote the paper, prepared figures and/or tables.
- Tatiana Vallaeys conceived and designed the experiments, analyzed the data, reviewed drafts of the paper.
- Ayi Santika and Ewa Borzym performed the experiments, contributed reagents/materials/analysis tools, reviewed drafts of the paper.
- Philippe Leleux analyzed the data, prepared figures and/or tables, reviewed drafts of the paper.
- Christophe Klopp analyzed the data, contributed reagents/materials/analysis tools, reviewed drafts of the paper.
- Jean-Christophe Avarre conceived and designed the experiments, performed the experiments, analyzed the data, contributed reagents/materials/analysis tools, wrote the paper, prepared figures and/or tables.

## DNA Deposition

The following information was supplied regarding the deposition of DNA sequences:

GenBank: KX544842, KX544843, KX544844, KX544845, KX544846, KX544847, KX544848.

## Data Availability

Sequence read Archive: SRP study accession #SRP059764; http://trace.ncbi.nlm.nih.gov/Traces/sra/?study=SRP059764.

## Supplemental Information

Supplemental information for this article can be found online at http://dx.doi.org/10.7717/peerj.2516#supplemental-information.

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
