# Peer review of "Targeted genomic enrichment and sequencing of CyHV-3 from carp tissues confirms low nucleotide diversity and mixed genotype infections"

_PeerJ, doi:10.7717/peerj.2516_

## Round 0.1 · original submission · Minor Revisions

The reviewers agreed on the scientific quality of the paper but both consider that a number of slight modifications should be done.

Suggestions are numerous, but need to be addressed before the paper can be considered as acceptable for publication.

·

Basic reporting

Raw data has been deposited in the SRA and, as of the time of this review, is publicly accessible. However, the consensus assemblies are not currently available and no discussion as the quality of those assemblies has been presented. However, as these assembles are not covered in any great detail in the body of the manuscript I do not feel this is a problem.

The article in question confirms to all PeerJ policies and is written in clear unambiguous English. I have provided an annotated PDF (notes should be visible in Foxit PDF reader) with corrections to the manuscript which includes some minor corrections and all of the points discussed below.

Page 4, line 19 – Please include a reference or details for the LM-PCR protocol used.

Page 4 40 - “Sequences were stored in the NG6 repository (Mariette et al. 2012) of GenoToul (Toulouse, France).” - This repository is not open access so it would be better practices to include the information for the raw reads from the SRA here (included later in the document) and remove the reference to GenoToul.

Page 4, Line 41 – Please detail the read quality control so the filtering step can be reproduced.

Page 4, Line 50 – Was the GATK used for duplicate read removal? If so please include this in this section.

Page 5, Line 5 – Please detail the read trimming parameters used for the sake of reproducibility and reference FastQ Trimmer.

Page 5, Line 16 - “Reads were artificially created for the 3 reference genomes U, I and J in order to include them in the analyses.” - Please provide information on how you accomplished this for reproducibility.

Table 2 – Perhaps include the 'Estimated number of viral copies' column from Table 1 here for easy visual comparison of the correlation between # raw reads and viral copy number.

Table 3 – The term 'variations' is vague. Could a more explicit term be used such as % small nucleotide variants (SNVs). Similarly “% intra-specimen variations” could be “% intra-specimen heterogeneous sites”. It is currently very easy to confuse the two columns. The same holds true for terminology in Figure 2 and 4.

Figure 1 – Axis tick marks would be useful for interpreting this figure.

Figure 1S – Panel B needs an axis title on the Y-axis. Tick marks may help interpretation.

Figure 2 – The term variations is vague/undefined.

Figure 3 – suggestion - include all samples of PoB3 in dendrogram (see below).

Figure 4 – Include window size or smoothing method used in the figure legend.

Experimental design

The study represents a comprehensive and rigorous analysis of the methodology presented. Sufficient controls have been included to identify methodologically introduced variants between samples.

Some small points on the methodology that should be addressed are included below:

Page 4, Line 41 - The raw read information contains a very, small amount of Illumina Universal adapter. It is not clear if reads were trimmed during the QC mentioned in this section. The presence of Illumina adapter should not influence the mapping in a meaningful way unless there was adapter content in the reference genome.

Page 5, Line 5 – Out of interest did you subsample your reads before assembly or use extremely high coverage read data for the assemblies? Did this have an effect on the assemblies?

Page 5, Line 10 - “Consensus sequences were generated from these assemblies”. Please clarify what you mean here - do you mean that the contigs that mapped to DQ657948 were used to represent the KHV genome that you recovered?

Page 5, Line 10 - The assemblies do not seem to have been used to produce any of the major results from the paper. I suggest you include a supplementary figure of the phylogeny generated from the assembly alignments for comparison to Figure 3 (see below).

Page 5, Line 11 - “Indels and SNPs were validated when more than 50% of the reads harbored the variation.” Did you map you reads back to your assemblies or otherwise compare your contigs to the reference? In case of the latter please could you provide more detail to enable reproducibility. Apologies if this is a step that is involved in Cap3 assembly, I have never use this software.

Page 5, Line 19 – Were any non-default parameters passed to hclust or PLINK?

Page 6, Line 43 - “Multiple sequence alignment realized on the seven generated consensus genomes and the three reference CyHV-3 genomes led to the same result.” - Could this figure (and the associated methodology) be included as a supplementary figure to allow for comparisons to be made to Figure 3.

Validity of the findings

Page 1, line 21 and Abstract - The abstract concludes with the statement “this new method will be valuable to trace back outbreaks in a quick way and at a reasonable cost”. The authors have not contrasted the cost or time frame of the current methodology to currently available alternatives within the text. A very basic analysis of the cost of single sample, duplex and and 4x multiplexed samples would be of interest to readers and would be needed to justify the inclusion of this line in the abstract.

Page 5, Line 26 - I could not make any assessments of the assemblies due to the lack of an accession number. A brief mention about the quality of the assemblies and some summary statistics would be useful. Perhaps in the supplemental materials.

Page 5, Line 34 - “The rate of enrichment, reflected by the proportion of
mapped reads, apparently correlated with the initial viral load, as it tended to increase concomitantly” - Was this significant? (I assume so).

Page 5, Line 40 – Figure 1S - The pattern of read coverage within and between samples looks very similar (i.e. similar position of peaks and troughs). Might this reflect improved recovery by some probes over others or is it perhaps an artefact of the mapping? The second alternative can be refuted by mapping simulated reference reads back to the reference genome and identifying if there are any regions with irregular coverage (the assumption would be you would have perfectly regular coverage with simulated data). This would be worthwhile to investigate and very quick to test.

Page 6, Line 42 – Figure 3 – Could the methodological replicates of PoB3 be included in this figure. This would give the reader a way of visually assessing how the between introduced variation caused by single sample, double capture or 4x multiplexing may impact on the distance measurements between related or identical samples. This would allow the reader to see the practical impact of the contamination mentioned on Page 7, Line 22 would have on sample differentiation (even, or perhaps especially, if it is effectively nil).

Page 7, Line 44 – Side note: Do you have any suggestions on how to identify 'real' heterogeneous sites relative to false ones without replicating samples? From sample PoB3 can you estimate how many (proportion) might be true real?

Page 7, Line 44 – Did the no of mapped reads after de-duplication correlate with the number of observed heterogeneous sites?

Page 8 – Line 11 – This maybe outside of the purview of this work but it would be an interesting to investigate. Was any variation in coverage observed at the VNTR loci in your samples before or after the read deduplication step that may be correlated with VNTR number?

Page 8, Line 24 - As previously mentioned some discussion of the costs of relative methods would be worthwhile. How dramatically does it impact on cost per sample?

Additional comments

This article was a very well rounded and enjoyable read. I hope my comments help to improve the manuscript for publication. Many of the points are minor but need to be addressed if this work is to be considered reproducible by future researchers. The majority of the other comments are speculation on my part or points relating to ease of interpretation by the reader.

Reviewer 2 ·

Basic reporting

The ms largely fulfils the basic reporting criteria apart from poor grammar in a few sections. Also, some of the figures are not appropriately labelled.

Experimental design

The ms largely fulfils the experimental design criteria. The experimental design is sound and methods have been carried out appropriately. However, in the methods section, more details should be provided on the various parameters used for the bioinformatics tools, as well as software version numbers and references for these tools.

Validity of the findings

The findings appear to be valid.

Additional comments

This manuscript (ms) describes a targeted enrichment protocol that enables sequencing of full-length CyHV-3 genomes directly from infected fish tissues. The developed genome capture system greatly reduces the cost of CyHV-3 whole genome sequencing and facilitates high-throughput analysis of infected samples, which could be used for investigations into new outbreaks.

Specific comments on the text.
Abstract page 2
L13: suggest ‘to recover’ is replaced with ‘recovery of’.
L22: replace ‘in a quick way’ with ‘rapidly’.
L23: replace ‘spreading routes of CyHV-3.’ with ‘routes of spread of CyHV-3.’

Introduction page 3
L4: replace ‘carps’ with ‘carp’ and ‘Since its…’ with ‘Since the…’.
L5-6: replace ‘it has spread in…’ with ‘the disease has spread to…’ and ‘Its causative agent, the cyprinid herpesvirus 3…’ with ‘The causative agent, cyprinid herpesvirus 3…’.
L11: Replace ‘It’ with ‘Although’ and delete ‘still’.
L12: Replace first use of ‘its’ with ‘the’ and delete the second ‘its’.
L14: Delete ‘one to another’.
L19: Insert ‘et al.’ after ‘Kurita’.
L23: replace ‘carps’ with ‘carp’.
L31-32: should read ‘enabling the sequencing of CyHV-3…’.

Materials and Methods, Page 4
L34: should read ‘consisted of…’.
L42: reference needed for BWA
L46: what parameters were used for the variant calling using GATK?
L49: INDELS should be in lowercase
Page 5
L3: add "NCBI": ...NCBI nucleotide collection
L5: FastQ should be all in lowercase (fastq) and provide version and reference for fastq trimmer.
L7: Provide version, reference and parameters used for CAP3.
Details should be added to the M&M section on how the duplicate reads were removed and on how the Venn diagram (Figure 2) and Circos plot (Figure 4) were generated, including version numbers of the tools and appropriate references.

Results
Page 5, Lines 43 & 47: ‘over representation’ should be two words.
Discussion
Page 6, lines 48-49: should read ‘ to recover ten complete (or near-complete) 295-kbp genomes…’.
Page 7, L3: remove "If" and lines 3-7: This long sentence is not well structured and needs to be rewritten (e.g. two sentences with second sentence beginning ‘However, there are a limited number of studies describing pre-capture multiplexing….’).
Page 7, lines 17 & 41: replace ‘4’ with ‘four’.
Page 8, line 1: replace ‘carps’ with ‘carp’.
Page 8, line 18-19: should read ‘..enable the study of the variations…’.
The authors discussed that some regions of the CyHV-3 genome are similar to regions in the host genome (e.g. P5,L45-46 & P6, L1-2). Based on this finding, it is most likely that the SNPs detected in these regions represent differences in sequences between host and pathogen. How many SNPs were detected in those regions? Judging from the Circos plot, there are likely to be many. Also, do the differences in heterozygous SNPs observed in these regions explain the difference in the number of SNPs for the 4 PoB3 samples? All of this needs to be discussed in the manuscript.

Tables & Figures
P14, Table 3, line 1: ‘SNPs’ and ‘indels’ should be written in full, followed by the (abbreviation) in the table title.
The figures support the main text and are relevant, but more details are needed. For example, the Circos plot (Figure 4) is difficult to interpret as important labels are missing (axes/units) and the plot needs to be explained in greater detail (how are the indels/SNPs represented?). There is a typo in the legend of Figure 3: "dendogram" should be replaced with "dendrogram". Figure 1S is missing a label on the y-axis. Some methods/approaches are only mentioned in the legends of these figures and this information should also be included in the methods section.

---

## Round 0.2 · Minor Revisions

Your manuscript was sent back to the former referees and one of them added one comment and suggested minor corrections. Please submit a final version that takes them into account before your manuscript can be accepted for publication.

·

Basic reporting

The standard of reporting used in the manuscript has greatly improved since the last review. A number of small revisions and additions are still required. See below.

Line 141 - A reference for FastQC is needed here.

Line 160 – 'Briefly, fastq files were trimmed using Fastq trimmer' – what parameters were used to trim reads i.e. what quality or length thresholds were used to trim the reads.

Line 298 - Consider substituting “Of these 75 variants, only 10 were located in the region that shares a high sequence identity with the carp reference genome.” for “Besides, only 10 of them were located in the region that shares a high sequence identity with the carp genome”.

Line 299 – Consider substituting “This evidence is likely indicative of low level cross-contamination with co-enriched samples.” for “All this tends to indicate that a “contamination” may have occurred during the enrichment procedure.”

Line 306 – Make this line clearer. Consider substituting “assignment of reads to sample during demultiplexing.” or something similar for “read affectation”.

Line 326 – Remove the period after iii)

Line 345 – Substitute “transmission routes” for “spreading routes”.

Experimental design

No Comments

Validity of the findings

The authors have addressed the majority of the previous concerns I had with the validity of the finding of the manuscript. They have highlighted potential shortfalls of their analysis and analysed problematic areas of the methodology. I only have one comment:

Line 346 – “Therefore, full genomic comparisons will probably be more suitable to trace back outbreaks and to understand the spreading routes of CyHV-3, as they will also enable the study of the variations of all tandem repeats contained in the genomes of CyHV-3.”. Large tandem repeats are notoriously difficult to study using short read sequencing so you may want to temper this sentence somewhat. See http://www.ncbi.nlm.nih.gov/pmc/articles/PMC3324860/ for examples of the problems caused by repeat regions. In the case of De Bruijn graph assemblers repeat units of length greater than the k-mer are problematic. Similarly, repeats greater than the length of the read are problematic for OLC assemblers. In both situations repeats will cause ambiguities or misassemblies. Similarly mapping of repeats and identification of the real copy number from mapping based approaches can be difficult (see http://www.ncbi.nlm.nih.gov/pubmed/25644268 for an example of an algorithm suitable for investigating this). You might consider removing the mention of tandem repeats in this sentence or replacing “all tandem repeats” with “short tandem repeats” to alleviate my concerns about this sentence.

Additional comments

The manuscript has been much improved. It is a comprehensive assessment of the methodology that could prove of great utility for studying the spread of KHVD and other viral pathogens in a cost effective manner.

---

## Round 0.3 · accepted · Accept

The paper is now acceptable for publication.